# NLPBench: Evaluating Large Language Models on Solving NLP Problems

## Abstract

Recent developments in large language models (LLMs) have shown promise in enhancing the capabilities of natural language processing (NLP). Despite these successes, there remains a dearth of research dedicated to the NLP problem-solving abilities of LLMs. To fill the gap in this area, we present a unique benchmarking dataset, NLPBench[1], comprising 378 college-level NLP questions spanning various NLP topics sourced from some universitys' prior final exams in the last decade, collected by professors and over 30 TAs. NLPBench includes questions with context, in which multiple sub-questions share the same public information, and diverse question types, including multiple choice, short answer, and math. Our evaluation, centered on LLMs such as GPT-3.5/4, PaLM-2, and LLAMA-2, incorporates advanced prompting strategies such as chain of thought (CoT) and tree of thought (ToT). Our study reveals that the effectiveness of the advanced prompting strategies can be inconsistent, occasionally damaging LLM performance, especially in smaller models like the LLAMA-2 (13b). Furthermore, our manual assessment illuminated specific shortcomings in LLMs' scientific problem-solving skills, with weaknesses in logical decomposition and reasoning notably affecting results.

## 1 Introduction

Over the past decade, the evolution of natural language processing (NLP) has led to the emergence of large language models (LLMs) (Brown et al., 2020; OpenAI., 2022; 2023; Zhang et al., 2023b; Touvron et al., 2023a; Zhang et al., 2023a; Gao et al., 2023b; Liu et al., 2023; Gao et al., 2023a). They consistently showcase exceptional performance across a spectrum of benchmarks that require human-level problem-solving or question-answering skills, including areas such as algebra (Lu et al., 2022; 2021b; 2023a; Cobbe et al., 2021), logic (Zhong et al., 2023; Chen et al., 2023), language (Huang et al., 2023), and science (Wang et al., 2023), some of these even challenges for well-educated individuals. As the most notable achievement in the field of NLP, a compelling, yet unresolved question of LLMs naturally arises: Can LLMs accurately answer questions about NLP?

To fill the gap in evaluating LLMs on NLP-related topics, we introduce a novel benchmark, **N**atural **L**anguage **P**rocessing **Bench**mark, referred to as NLPBench. Our NLPBench contains 378 high-quality NLP-related questions from a university's final exams in the last decade. Collected questions are in the fields of *Language Modeling and Syntax Parsing*, *Semantics and Logic*, *Pragmatics, Discourse, Dialogue and Applications*, *Information Retrieval and Topic Modeling*, *Artificial Intelligence* and *Other Topics*. To evaluate the multi-turn communication problem-solving ability of different NLP topics, we introduce questions with context, consisting of multiple related questions that share the same public information. Our dataset also includes multiple choice, free response short answer, and math questions to evaluate LLMs from all perspectives. Figure 1 shows some example questions featured in our dataset.

We direct our evaluation towards five representative LLMs, GPT-3.5/4 (OpenAI., 2022; 2023), PaLM-2 (Anil et al., 2023), and both the 13b and 70b versions of LLAMA-2 (Touvron et al., 2023b). Our study incorporates a variety of advanced prompting strategies, including chain-of-thought (CoT, Wei et al. (2022)) and tree-of-thought (ToT, Yao et al. (2023)), and the argumentation method like

---

[1] https://anonymous.4open.science/r/NLPB-04A3

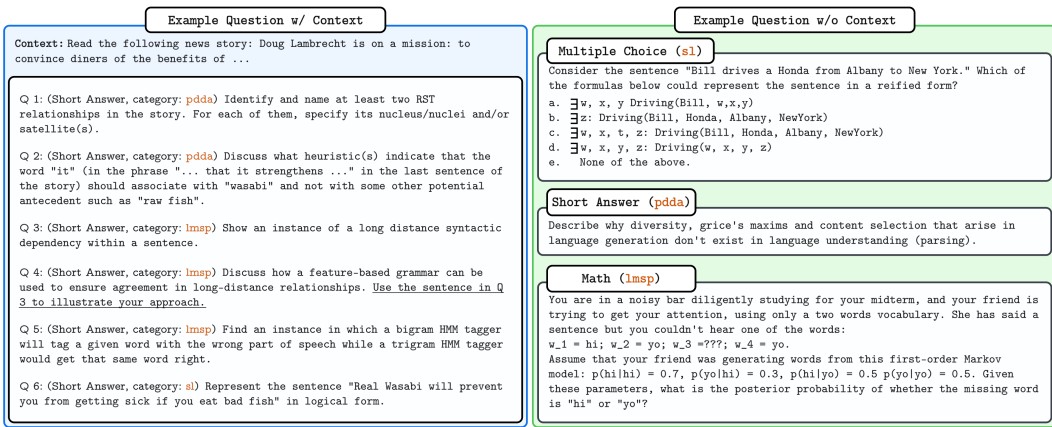

**Figure 1:** Example questions in NLPBench dataset. We collected three types of questions, including multiple choice, short answer, and math, and divided them into two categories: with and without context. Text underline shows the relations between questions.

self-consistency. These advanced prompting strategies have demonstrated notable success in past benchmarks by directing the LLMs' response processes. They guide LLMs with specific examples, encouraging the generation of step-by-step solutions that lead to deeper problem consideration (Wei et al., 2022; Wang et al., 2022; Zhou et al., 2022; Huang et al., 2022). However, the efficacy of these improvements can be compromised by the complexity of the question, the depth of required knowledge, and the LLMs' ability to follow prompts. Our experiments indicate that few-shot prompting typically results in modest enhancements. Moreover, advanced prompting strategies are not universally effective. When an LLM is constrained (for instance, by having insufficient parameters to develop a robust representation) or when the breadth of required knowledge expands, the LLM might not always recall accurate information from its previously stored knowledge. In our research, we observe that advanced prompting strategies can inadvertently hamper the performance of LLMs. This is due to the introduction of extraneous noise unrelated to the given questions, sometimes causing a pronounced decline in the performance of smaller LLMs, such as LLAMA-2 (13b). Such nuances have remained unexplored in earlier benchmarks because of the limited scope of question complexity and prompt length.

Apart from examining the effectiveness of various prompting strategies, we also conducted a manual assessment of NLP problem-solving capabilities in two dimensions: (1) error rate statistics across different NLP categories and (2) an evaluation of problem-solving abilities from a human expert's viewpoint. For the first dimension, we compiled the error rates for each NLP category, segmented by individual LLMs and their associated prompting strategies. Our findings indicate that few-shot prompts can decrease the error rate for specific question types by introducing domain-specific supplementary information. In contrast, other methods might not bring about a substantial reduction in error rates. For the second evaluation dimension, we initially identified seven scientific problem-solving skills. We then categorized the mistakes made by the LLMs to highlight deficiencies in these pre-established skills. Our findings underscore that the absence of skills in logical decomposition, problem deduction, and logical reasoning predominantly contributes to the subpar performance observed in our NLPBench. Based on the above evaluations, we conclude that simple prompting methods are enough for promising results, and the training process should focus more on fostering specific problem-solving skills like logical decomposition and reasoning.

## 2 THE NLPBENCH DATASET

We collect a new dataset consisting of final exam questions from the universities' NLP courses to evaluate the capabilities and analysis of the limitations of the existing large language models (LLMs) to solve NLP-related problems. All questions are divided into two types: with and without context, where a question with context consists of multiple related sub-questions sharing the same public information. Questions with context require answering with multi-turn communication. We further

**Table 1:** Statistic of the original dataset and the percent of usage in our proposed dataset.

| Categories | Short Answer | | Multiple Choice | | Math | |
|---|---|---|---|---|---|---|
| | w/ context | w/o context | w/ context | w/o context | w/ context | w/o context |
| # Total | 237 | 148 | 16 | 162 | 28 | 15 |
| % Answer | 67.1% (159) | 58.1% (86) | 93.7% (15) | 88.9% (144) | 92.8% (26) | 46.6/% (7) |
| % Used | 72.6% (130) | 48.4% (62) | 93.7% (15) | 88.9% (144) | 85.7% (24) | 20% (3) |

categorize each question according to the answer format: short answer, multiple choice, and math. This section introduces the details of the dataset construction process.

**Data selection.**   Initially, we amassed a substantial collection of approximately 1,000 NLP exam questions over a decade, from 2013 to 2023. Professors and teaching assistants (TAs) contributed new questions to this repository each semester. This extensive set comprises three types of questions: 1) Online-sourced questions, refined by TAs to differentiate them from their original versions, ensuring their uniqueness for final exams, and a thorough verification of answers. 2) Original questions formulated by professors, drawing from their teaching experience. 3) Original questions developed by TAs. Over 30 TAs have been involved in the completion of this dataset. The questions are of high quality and are not available online, maintaining the integrity and fairness of the final exams. We discard the questions with figures or tables, and the remaining 372 questions are used in NLP-Bench. Different from the previous benchmarks, our dataset introduces a new category *with context*, as shown in Figure 1, which requires more complex reasoning steps to capture the relation between the current question and context and the relation between current and other questions. Considering the evaluation of the basic ability of LLMs, our dataset also contains traditional *without context* questions. All of the above questions are further divided into multiple-choice, short answer, and math according to their answer type. Specifically, our proposed dataset has the following features:

- **Inclusion of NLP-related problems.**  The chosen problems demand a solid understanding of NLP-related knowledge (e.g., rhetorical structure theory, formal languages, application of probabilistic theory in NLP, etc.) in reasoning capability, the adaptation of calculation skills, and the ability to comprehend complex concepts.

- **Inclusion of detailed solutions**: To facilitate a thorough analysis of the limitations of LLMs, detailed solutions should be provided for the selected problems. This enables a comprehensive examination of the performance of LLMs and their capacity to handle complex problem-solving tasks.

- **Inaccessibility.** To ensure an unbiased evaluation, we carefully curate questions that are not readily accessible online and couldn't be easily extracted or transformed into text. This selection process aims to mitigate any potential information leakage from the exposure of LLMs to pre-existing online question banks, such as those found in standardized tests like the SAT exams.

- **Complex structure.**  About half of our collected questions have a complex structure, with a context shared with multiple subsequent questions and relations between each question. This type of question requires the model to solve with a multi-turn conversation and examine the model's ability to capture critical information in the context.

**Data processing.**   All questions are initially available in both text and image formats (e.g., handwritten), which we meticulously converted into plain text and LaTeX documents using a web-based annotation tool, and the extracted questions will be saved in JSON format. A detailed overview of the tool's user interface can be found in Appendix B. Expert human annotators rigorously reviewed each problem to guarantee the absence of LaTeX syntax errors and to ensure all characters adhere to the ASCII standard. We classified the questions into three formats: short answers, multiple choice, and mathematical. Furthermore, based on the inclusion or exclusion of context information, information common to a set of subsequent questions (e.g., paragraphs from a book, upon which the answers to all following questions are contingent), we divided the questions into two main categories: with and without context. Notably, we integrated the true-false format from the original dataset into the multiple-choice category due to its limited amount. Each question comes with a ground-truth answer for evaluation. Our dataset also contains short answer questions that require free-form responses, such as prompting for examples or specific subsets of a concept. This further reduces the chances of candidates simply guessing correct answers rather than only using multiple

choice questions (Lu et al., 2021a; 2022; Chen et al., 2023). To assist in evaluating responses to these questions, we offer sample answers that guide evaluators in determining the accuracy of a response. For mathematical problems, we document answers in LaTeX format, specifying exact figures, accompanied by their respective step-by-step solutions. These stepwise solutions serve as guides for intermediate reasoning methodologies (e.g., the "Chain of Thought" approach), assisting LLMs in formulating more accurate answers.

**Dataset statistics.** In summary, we collected 378 questions from some Universities' NLP course final exams. The dataset includes 192 short-answer questions, 159 multiple-choice questions, and 27 math questions with step-by-step solutions. All types of questions are divided into with context and without. We detailed the statistical results of each question type in Table 1. All questions were also orig-

**Table 2:** The question quantity under each NLP concept. All the categories are defined by human experts.

| Category | Acronym | # Questions |
|---|---|---|
| Language Modeling and Syntax Parsing | lmsp | 162 |
| Semantics and Logic | sl | 69 |
| Pragmatics, Discourse, Dialogue and Applications | pdda | 13 |
| Information Retrieval and Topic Modeling | irtm | 27 |
| Artificial Intelligence | ai | 75 |
| Other Topics | ot | 32 |

inally categorized into six common NLP-related concepts, summarized in Table 2. Specifically, the questions belong to *Other topics* are in the field of current research, speech processing, ethics, and applications to other domains.

## 3 EXPERIMENT

### 3.1 EXPERIMENT SETUP

We evaluate both the online accessible models (GPT-3.5, OpenAI. (2022), GPT-4, OpenAI. (2023) and PaLM-2, Anil et al. (2023)) and open-sourced models (LLAMA-2 (13 and 70b), Touvron et al. (2023b)) on the proposed dataset. We consider two advanced prompting strategies, including chain-of-thought (CoT, Wei et al. (2022)) and tree-of-thought (ToT, Yao et al. (2023)), under both zero-shot and few-shot with or without system prompt. We also perform self-consistency (SC) as an improvement of greedy methods.

- **Zero-shot and few-shot prompting.** Under zero-shot prompting, the model is not able to access questions in the training set for prior knowledge, which evaluates their inherent problem-solving capabilities with background knowledge and reasoning abilities. While in the few-shot prompting, a few examples are mixed into the input prompt as the prerequisites for the later questions. This aims to examine their capability to learn new information from the demonstrations and incorporate it into their problem-solving processes.

- **Advanced prompting strategies.** We try different prompting methods, zero-shot and few-shot, and we further combine them with or without system prompt, CoT, and ToT. We implement CoT in two ways: the traditional 2-staged (adding *let's think step by step* behind the questions) for short answer questions and format template for multiple choice and math questions. This is because of the hardness of extracting the reasoning chain from the short answer questions, different from the multiple choice and math, in which we can extract an exact problem-solving process easily by separating the final answer and the corresponding process.

In summary, we consider ten combinations of prompting strategies: zero-shot and few-shot prompting (*ZS*, *FS*), zero-shot and few-shot prompting with system prompt (*ZS+SYS*, *FS+SYS*), chain-of-thought prompting under zero-shot and few-shot (*ZS+CoT*, *FS+CoT*), chain-of-thought prompting under zero-shot and few-shot with system prompt (*ZS+CoT+SYS*, *FS+CoT+SYS*), and tree-of-thought under zero-shot and few-shot (*ZS+ToT*, *FS+ToT*). Zero-shot, few-shot, and CoT, with SC, are evaluated on the multiple choice question set due to the limitation of the statistic method in SC. Example prompts of the above method are provided in Appendix A.3.

**Implementation details.** We access the API of GPT-3.5 (`gpt-3.5-turbo`) and GPT-4 (`gpt-4`) via AutoGen[2] (Wu et al., 2023), which provided the enclosure of Open-AI API, helping us cache the results with same hyperparameters. We access PaLM-2 via the Google PaLM `generate_text`

---

[2] https://microsoft.github.io/autogen/

**Table 3:** Experimental results in terms of accuracy (%) on our proposed dataset. The best average scores in each type of question are highlighted in **bold in red**, and the best average scores for each model in a specific type of question are underlined in blue. Results marked with - denote the incomplete experiment caused by exceeding context length or other prompting errors.

| Model | Setting | | Multiple Choice | | | Short Answer | | | Math | | | Overall Acc. |
|---|---|---|---|---|---|---|---|---|---|---|---|---|
| | | | w/ Context | w/o Context | Average | w/ Context | w/o Context | Average | w/ Context | w/o Context | Average | |
| LLAMA-2 (13b) | ZS | Orig. | 20.00 | 20.83 | 20.75 | 39.23 | 37.10 | 38.54 | 20.00 | 0.00 | 4.00 | 28.72 |
| | | +SYS | 26.67 | 34.03 | 33.33 | 43.85 | 27.42 | 38.54 | 0.00 | 0.00 | 0.00 | 33.77 |
| | | +CoT | 26.67 | 19.44 | 20.13 | 22.31 | 9.68 | 18.23 | 0.00 | 0.00 | 0.00 | 17.82 |
| | | +CoT+SYS | 33.33 | 27.08 | 27.67 | 23.08 | 9.68 | 18.75 | 0.00 | 0.00 | 0.00 | 21.28 |
| | FS | Orig. | - | 31.25 | 28.30 | - | 29.03 | 9.38 | - | - | 0.00 | 16.76 |
| | | +SYS | - | 38.19 | 34.59 | - | 30.65 | 9.90 | - | - | 0.00 | 19.68 |
| | | +CoT | - | 30.56 | 27.67 | - | 32.26 | 10.42 | - | - | 0.00 | 17.02 |
| | | +CoT+SYS | - | 36.81 | 33.33 | - | 35.48 | 11.46 | - | - | 0.00 | 19.95 |
| LLAMA-2 (70b) | ZS | Orig. | 40.00 | 22.22 | 23.90 | 53.85 | 38.71 | 48.96 | 9.09 | 0.00 | 8.00 | 35.64 |
| | | +SYS | 40.00 | 23.61 | 25.16 | 54.62 | 46.77 | 52.08 | 9.09 | 0.00 | 8.00 | 37.77 |
| | | +CoT | 33.33 | 21.53 | 22.64 | 32.31 | 12.90 | 26.04 | 0.00 | 0.00 | 0.00 | 22.87 |
| | | +CoT+SYS | 40.00 | 38.19 | 38.36 | 33.08 | 25.81 | 30.73 | 0.00 | 0.00 | 0.00 | 31.91 |
| | FS | Orig. | 33.33 | 29.17 | 29.56 | 48.46 | 38.71 | 45.31 | 9.09 | 0.00 | 19.38 | 36.93 |
| | | +SYS | 26.67 | 34.72 | 33.96 | 46.92 | 40.32 | 44.79 | 0.00 | 0.00 | 0.00 | 37.23 |
| | | +CoT | 26.67 | 31.94 | 31.45 | 38.46 | 51.61 | 42.71 | 0.00 | 0.00 | 0.00 | 35.11 |
| | | +CoT+SYS | 26.67 | 38.19 | 37.11 | 35.38 | 48.39 | 39.58 | 4.55 | 0.00 | 4.00 | 36.17 |
| PaLM-2 | ZS | Orig. | 66.67 | 37.50 | 40.25 | 49.23 | 35.48 | 44.79 | 13.64 | 33.33 | 16.00 | 40.96 |
| | | +SYS | 66.67 | 45.83 | 47.80 | 51.54 | 37.10 | 46.88 | 4.55 | 33.33 | 8.00 | 44.68 |
| | | +CoT | 60.00 | 36.81 | 38.99 | 47.69 | 37.10 | 44.27 | 18.18 | 33.33 | 20.00 | 40.42 |
| | | +CoT+SYS | 53.33 | 41.67 | 42.77 | 40.00 | 30.65 | 36.98 | 13.64 | 0.00 | 12.00 | 37.77 |
| | | +ToT | - | 4.86 | 4.40 | - | 0.00 | 0.00 | - | - | 0.00 | 1.86 |
| | FS | Orig. | 53.33 | 38.89 | 40.25 | 57.69 | 33.87 | 50.00 | 4.55 | 33.33 | 8.00 | 43.08 |
| | | +SYS | 53.33 | 39.58 | 40.88 | 56.15 | 38.71 | 50.52 | 4.55 | 0.00 | 4.00 | 43.35 |
| | | +CoT | 53.33 | 40.28 | 41.51 | 49.23 | 38.71 | 45.83 | 0.00 | 0.00 | 0.00 | 40.96 |
| | | +CoT+SYS | 40.00 | 38.89 | 38.99 | 53.85 | 40.32 | 49.48 | 0.00 | 0.00 | 0.00 | 41.75 |
| | | +ToT | - | 10.42 | 9.43 | - | 1.61 | 0.52 | - | - | 0.00 | 4.25 |
| GPT-3.5 | ZS | Orig. | 40.00 | 52.05 | 50.64 | 75.38 | 58.73 | 69.99 | 36.36 | 33.33 | 36.00 | 59.55 |
| | | +SYS | 46.67 | 40.69 | 41.51 | 71.54 | 62.90 | 68.75 | 13.64 | 33.33 | 16.00 | 53.72 |
| | | +CoT | 53.33 | 52.74 | 52.83 | 63.85 | 33.33 | 54.19 | 18.18 | 100.00 | 28.00 | 51.87 |
| | | +CoT+SYS | 46.67 | 39.58 | 40.25 | 66.92 | 59.68 | 64.58 | 18.18 | 0.00 | 16.00 | 51.06 |
| | | +ToT | - | 31.25 | 28.30 | - | 0.00 | 0.00 | - | - | 0.00 | 11.97 |
| | FS | Orig. | 53.33 | 36.81 | 38.36 | 66.15 | 64.52 | 65.62 | 18.18 | 33.33 | 20.00 | 51.06 |
| | | +SYS | 46.67 | 44.44 | 44.65 | 66.15 | 54.84 | 62.50 | 18.18 | 0.00 | 16.00 | 51.86 |
| | | +CoT | 40.00 | 40.28 | 40.25 | 64.62 | 62.90 | 64.06 | 13.64 | 0.00 | 12.00 | 50.53 |
| | | +CoT+SYS | 40.00 | 46.53 | 45.91 | 66.15 | 64.52 | 65.62 | 18.18 | 0.00 | 16.00 | 53.99 |
| | | +ToT | - | 30.56 | 27.67 | - | 56.45 | 18.23 | - | - | 0.00 | 21.01 |
| GPT-4 | ZS | Orig. | 86.67 | 70.55 | 72.25 | 78.46 | 69.84 | 75.42 | 22.73 | 33.33 | 24.00 | 70.66 |
| | | +SYS | 86.67 | 57.93 | 60.38 | 83.85 | 79.03 | 82.29 | 18.18 | 0.00 | 16.00 | 68.62 |
| | | +CoT | 86.67 | 72.60 | 74.10 | 74.62 | 57.14 | 68.65 | 13.64 | 100.00 | 24.00 | 67.99 |
| | | +CoT+SYS | 86.67 | 56.25 | 59.12 | 73.08 | 75.81 | 73.96 | 27.27 | 66.67 | 28.00 | 64.63 |
| | | +ToT | - | 60.42 | 54.72 | - | 0.00 | 0.00 | - | - | 0.00 | 23.14 |
| | FS | Orig. | 86.67 | 62.50 | 64.78 | 77.69 | 75.81 | 77.08 | 22.73 | 0.00 | 20.00 | 68.08 |
| | | +SYS | 86.67 | 59.03 | 61.64 | 81.54 | 79.03 | 80.73 | 13.64 | 33.33 | 16.00 | 68.35 |
| | | +CoT | 86.67 | 60.42 | 62.89 | 78.46 | 75.81 | 77.60 | 36.36 | 0.00 | 32.00 | 68.35 |
| | | +CoT+SYS | 86.67 | 60.42 | 62.89 | 80.00 | 74.19 | 78.12 | 13.64 | 66.67 | 20.00 | 67.82 |
| | | +ToT | - | 60.42 | 54.72 | - | 75.81 | 24.48 | - | - | 0.00 | 35.64 |

**Table 4:** Comparison of prompting methods with and without self-consistency (denoted as SC) on GPT-3.5, GPT-4, and PaLM-2. All results are statistics from the multiple-choice question set.

| Model | ZS | | ZS+CoT | | FS | | FS+CoT | |
|---|---|---|---|---|---|---|---|---|
| | w/o SC | w/ SC | w/o SC | w/ SC | w/o SC | w/ SC | w/o SC | w/ SC |
| GPT-3.5 | 50.64 | 37.11 | 52.83 | 38.36 | 38.36 | 43.40 | 40.25 | 44.03 |
| GPT-4 | 72.25 | 59.75 | 74.10 | 62.89 | 64.78 | 64.78 | 62.89 | 66.67 |
| PaLM-2 | 40.25 | 23.90 | 38.99 | 28.30 | 40.25 | 37.11 | 41.51 | 38.99 |

API[3], which is recommended by Google for problem-solving and handling zero and few shot tasks. For open-source models LLAMA-2 (13b and 70b), we use the endpoint implemented by vLLM[4] (Kwon et al., 2023), an open-sourced, fast-speed LLM serving platforms for a wide range of open-source models, which can provide Open-AI like API for the LLM user. We further access those endpoints via AutoGen, the same as we access the Open-AI model. For all models, we use the same seed and set the temperature as 1 for question answering and 0 for the middle process in CoT and ToT. We choose a high temperature for a more creative answer and a low temperature for a more specific process.

---

[3] https://developers.generativeai.google/products/palm
[4] https://vllm.readthedocs.io/en/latest/

## 3.2 RESULTS AND ANALYSIS

The experimental results for GPT-3.5, GPT-4, PaLM-2, and LLAMA-2 (13b and 70b) with various configurations on our NLPBench are detailed in Table 3. Supplementary analysis utilizing conventional text evaluation metrics, such as ROUGE-L and CIDEr, can be found in Appendix A.1. We highlight the model performance by presenting accuracy scores in both 'with' and 'without' context scenarios. Notably, questions requiring context involve multi-turn interactions with the model. Our accuracy calculation focuses on the model's **final answer**, disregarding intermediary steps when computing accuracy, which will be considered in the human evaluation process. For context-based questions, we examine the accuracy of each distinct sub-question. From the experiment results, we have several key observations:

**GPT-4 outperforms all models with a significant margin under most of the situations.** Based on the results across three distinct question formats categorized under two categories, GPT-4 outperforms all baselines under most situations. Specifically, it achieved the top spot with the best average performance accuracy in two of the question formats. When juxtaposed against all baseline methods, there's a remarkable uplift in its performance, registering an average score improvement of at most 67.85% and 82.29% when compared with LLAMA-2 (13b). It's worth highlighting that these outstanding results were obtained under a zero-shot setting without the aid of any sophisticated prompting strategies. Interestingly, our observations also indicate that deploying advanced prompting techniques often has a counterproductive effect on GPT-4's performance in many scenarios.

**Few-shot prompting does not always improve.** In Figure 2, we present a comparison of average performance between zero-shot and few-shot prompting. Notably, the adoption of few-shot prompting often results in a modest performance enhancement, and in some cases, even a decrease, consistent with findings by Wang et al. (2023). A closer examination of Table 3 reveals that in some cases, LLAMA-2 (13b and 70b) derives advantages from the supplementary knowledge gained through few-shot prompting. However, this can lead to surpassing the maximum context length, particularly when multi-turn communication is necessitated, or the query contains an extensive description, which leads to a significant performance drop in LLAMA-2 (13b). GPT-3.5, GPT-4, and PaLM-2 only have ordinary improvements, about 3%,

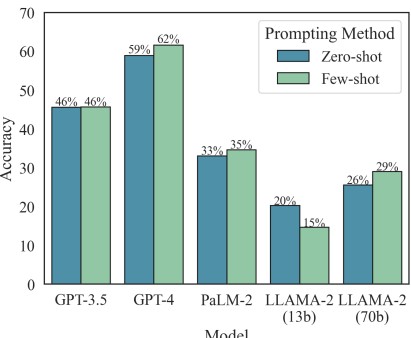

**Figure 2:** Zero-shot v.s. few-shot prompting on overall accuracy(%).

when adopting few-shot prompting. In fact, seven out of the nine highest average scores were realized using zero-shot prompting. This phenomenon may arise because the chosen sample questions are either highly representative of and specific to the domain or, conversely, do not capture its diversity adequately, introducing errors during inference. Therefore, while few-shot prompting can potentially extend the prompt length and occasionally enhance performance, the selection of sample questions is critical. Ill-chosen examples can introduce noise detrimental to the task at hand.

**Advanced prompting strategies do not work consistently, sometimes having a negative effect.** In Figure 3, we present the average scores both with and without the utilization of advanced prompting strategies. Notably, CoT only provides a slight performance increase with GPT-3.5 and will cause performance declines in other models. The efficacy of these prompting strategies is heavily dependent on the model's innate ability to adhere to the prompts, which necessitates the models to self-evaluate their responses. CoT demands a singular feedback loop, which is relatively straightforward. In contrast, ToT calls for multiple feedback mechanisms coupled with a search operation, such as the DFS algorithm. Challenges arise with ToT when a model generates a response that diverges from the specified template in the

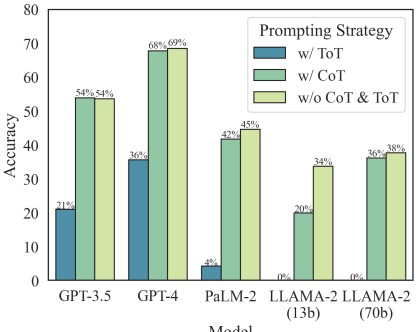

**Figure 3:** Overall accuracy(%) with and without advanced prompting strategies.

prompt. GPT-3.5/4 exhibits an exceptional capacity to process intricate prompts, yielding the SOTA results (when comparing with other models) in tasks that necessitate intricate logical reasoning when

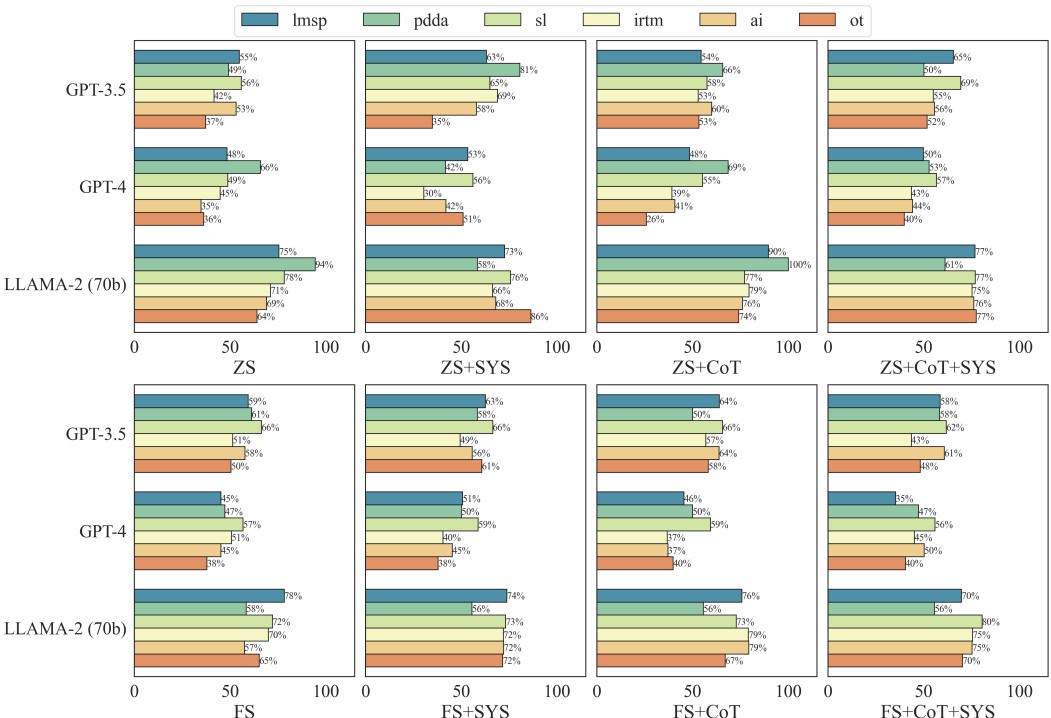

**Figure 4:** The comparison of overall **error rate**(%) between GPT-3.5/4 and LLAMA 2-70b across all prompting strategies of each NLP category. Each color bar indicates a pre-defined NLP category from the original dataset.

implementing advanced prompting strategies but still cannot outperform the baseline without any prompting strategy. While LLAMA-2 (13b), due to the limited prompt-following capability and constricted context length, it experienced a downturn in performance when employing these advanced strategies. On the other hand, self-consistency (Wang et al., 2022), a robust alternative to greedy decoding, demonstrates impressive results on other benchmarks. Nevertheless, our findings, detailed in Table 4, indicate that while self-consistency can enhance performance with few-shot prompting (as seen with GPT-3.5 and GPT-4), it considerably undermines the output during zero-shot prompting. A potential explanation for such contrasting outcomes is that few-shot prompting restricts the scope of knowledge, impacting answer generation, a constraint absent in zero-shot prompting.

## 4 ERROR ANALYSIS OF VARIOUS PROMPTING STRATEGIES

Considering the substantial advancements of current Large Language Models (LLMs), an in-depth analysis of the particular skills that are either enhanced or limited under certain settings becomes imperative. We evaluate two types of abilities that should be obtained before taking the final exam: an understanding of natural language processing (NLP) and the ability to solve college-level problems. We select the results provided by GPT-3.5/4 and LLAMA 2-70b, which represent the SOTA online and open-sourced model, respectively.

### 4.1 UNDERSTANDING OF NATURAL LANGUAGE PROCESSING

To assess the NLP comprehension of LLMs, we delineated the errors made by GPT-3.5/4 and LLAMA 2-70b in Figure 4, showcasing their respective error rates across various NLP categories. A notable disparity in distribution is evident between zero-shot and few-shot prompting. There's a marked decrease in error rates for `pdda` by 16% for GPT-4 and 32% for LLAMA 2-70b when transitioning from zero-shot to few-shot prompting, a trend similarly noted in the CoT results. However, this trend diminishes once a system prompt is integrated. The introduction of a system prompt and additional example questions helps mitigate errors stemming from incorrect prior knowledge. Yet, combining the system prompt with few-shot prompting increases the error rate by 10% on `irtm` and

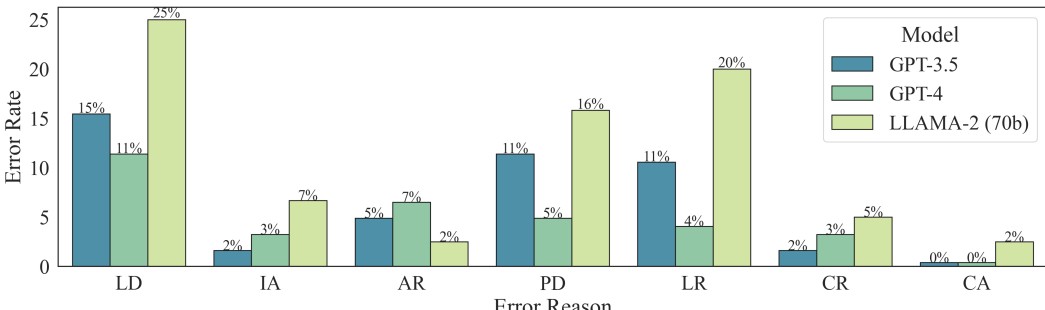

**Figure 5:** The error profiles of the deficient of seven essential science problem-solving abilities between GPT-3.5/4 and LLAMA 2-70b. The height of the color bars indicates the percentage that the model has an incorrect answer due to a lack of corresponding science problem-solving skills.

8% on `pdda` for GPT-4. In contrast, there's a 13% reduction in the error rate for `ot`. For LLAMA 2-70b, few-shot prompting consistently reduces error rates across categories, resulting in a more balanced error distribution.

In summary, few-shot prompting can help decrease the error rate for certain types of questions by offering additional examples from the dataset. However, its effectiveness diminishes when the dataset demands a broad spectrum of knowledge. While advanced prompting strategies like CoT may not substantially enhance performance with complex datasets, system prompts can counteract errors introduced by these advanced strategies.

## 4.2 ABILITY TO SOLVE COLLEGE-LEVEL PROBLEMS

We chose three models, both online and open-sourced, with the best average performance (GPT-3.5 w/ ZS, GPT-4 w/ ZS, and LLAMA 2-70b w/ ZS+SYS) and annotated the source of the error for short answers (with a unique answer) and math questions, indicating where the model made a mistake and why. Following Wang et al. (2023), we classify the human-annotated error reasons into seven crucial skills deficient for solving complex college-level problems. For each wrong question, we summarized three of the seven skills:

- **Logical decomposition and analysis (LD).** This ability involves decomposing the question into smaller, manageable parts and understanding the relationships between these parts.

- **Identification of assumptions (IA).** This skill involves the ability to recognize relevant and necessary assumptions in the question.

- **Causal reasoning (CR).** This is the ability to understand cause-and-effect relationships.

- **Problem deduction skills (PD).** This pertains to the ability to infer and deduce potential solutions or underlying principles from the given information in a problem.

- **Abstract reasoning (AR).** This skill involves the ability to understand complex concepts that cannot be perceived physically and to recognize patterns or relationships beyond concrete examples.

- **Logical reasoning (LR).** This is the ability to make a reasoned argument and to identify fallacies or inconsistencies in an argument or set of data.

- **Calculation (CA).** This involves the ability to carry out mathematical operations and computations accurately.

The analysis results are recorded in Figure 5, we also provided some error samples in Appendix A.2. Compared with the SOTA GPT-4, GPT-3.5 has 6% and 7% higher probability of making wrong answers caused by a lack of problem deduction and logical reasoning skills, and LLAMA 2-70b has 14%, 11%, and 16% higher in logical decomposition, problem deduction and logical reasoning skills. This increment reveals a strong relation between a correct answer and logical decomposition, problem deduction, and logical reasoning skills, which is similar to the findings of Berglund et al. (2023). Many questions in our NLPBench dataset require an understanding of a given text before the question (e.g., a story or news). Answer such questions need to retrieve the critical information in the context and build up a logical relation between the question and the retrieved information, which requires a high-level logical decomposition and logical reasoning ability. We also found that

GPT-3.5 and 4 do not lack calculation skills but have a low accuracy in math questions (see Table 3). This is because models need to understand the question before the calculation, and the question in our dataset is hard (e.g., requires an understanding of the EM algorithm). Therefore, models often give an answer that is correct in the calculation with a completely wrong process.

## 5 RELATED WORKS

Traditional benchmarks have been oriented toward assessing the general abilities of models. For instance, SQuAD (Rajpurkar et al., 2018) was developed to gauge models' reading comprehension skills. GLUE (Wang et al., 2018) provides a versatile framework for evaluating performance across a variety of natural language understanding tasks. Cosmos QA (Huang et al., 2019) delves into assessing models on their common-sense reasoning abilities using natural language contexts. HumanEval (Chen et al., 2021) targets the coding prowess of models, presenting 164 Python programming challenges. BIG-Bench (Srivastava et al., 2022) serves as a comprehensive test suite that includes 204 multiple-choice or exact-match tasks, while its counterpart, BIG-Bench Hard (Suzgun et al., 2022), presents notably intricate chain-of-thought prompts. Finally, HELM (Liang et al., 2022) offers a detailed multi-metric evaluation of LLMs, shedding light on their strengths, weaknesses, and potential risks.

Recent benchmarks predominantly assess LLMs' problem-solving skills, particularly in science and mathematics (Lu et al., 2023b; Fu et al., 2023; Lu et al., 2023a; Zhong et al., 2023; Mishra et al., 2022; Chen et al., 2023; Guo et al., 2023; Hendrycks et al., 2020). Noteworthy datasets include GSM8K (Cobbe et al., 2021), which contains 8.5K elementary math word problems, ScienceQA (Lu et al., 2022), a multimodal dataset with lectures, and MATH (Hendrycks et al., 2021), consisting of 12.5K problems from math contests. LILA (Mishra et al., 2022) enhances 20 datasets with task guidelines and Python solutions. Most benchmarks focus on foundational arithmetic, but TheoremQA (Chen et al., 2023) offers 800 theorem-centric questions. Galactica (Taylor et al., 2022) explores scientific tasks, such as latex equation conversions, while C-EVAL (Huang et al., 2023) evaluates LLMs within a Chinese cultural context. AGIEval (Zhong et al., 2023) measures LLM performance against standardized tests using human-annotated analysis. SciBench (Wang et al., 2023) presents college-level science problems from textbooks with an automatic evaluation method. However, while these benchmarks emphasize single-turn communication, ours assesses the multi-turn problem-solving capabilities of LLMs. A detailed comparison is provided in Appendix C.

## 6 CONCLUSION AND RECOMMENDATION

This study unveils NLPBench, a collection of 378 college-level NLP questions aimed at comprehensively evaluating Large Language Models (LLMs) like GPT-3.5, GPT-4, and others. NLPBench is designed for testing LLMs' proficiency in multi-turn conversations, using advanced prompting strategies such as chain-of-thought and few-shot prompting. However, the evaluation indicates that these strategies don't always enhance performance. A closer look at errors made by models like GPT-3.5/4 and LLAMA 2-70b suggests they mainly falter in logical deconstruction and reasoning, leading to their limited success on NLPBench. Based on the above conclusion, we have the following recommendations:

- **Simple Prompting method is enough for promising results.** Based on our findings in Section 3.2, we found that few-shot prompting averagely surpasses zero-shot, but it is hard to achieve the best. Section 4.1 indicates that while few-shot can decrease errors in certain categories, it can also lead to more verbose prompts. Employ few-shot prompting when your task is concentrated on a specific domain.

- **Advanced prompting strategies are not necessary.** They show weak or roughly comparable results to zero-shot on all LLMs and will significantly affect the relatively small LLM (e.g., LLAMA 2-13b). As described in Section 3.2, advanced prompting strategies need strong prompt follow-up ability, since they all require multiple reasoning steps. If budget is one of your limitations, zero-shot is also a good choice for a competitive result.

- **The pretraining process should focus more on fostering "logical thinking skills"** According to Section 4.2, we found that LLAMA 2 clearly lacks the ability to do logical decomposition, problem deduction, and logical reasoning. We believe that LLM training should take into account these three dimensions.

ETHICS STATEMENT

NLPBench aims to evaluate the NLP-related problem-solving ability of LLMs. Our evaluation results provide efficient insight for further research on advanced prompting strategies or LLM pre-training by testing and analyzing the reason for the errors made by LLMs. NLPBench does not contain any personal, sensitive, or confidential data and is diverse and representative of a wide range of scenarios, demographics, and contexts.

REPRODUCIBILITY STATEMENT

Our main experiments are done on online accessible or open-sourced models (except for PaLM-2, which is not an openly accessible model yet). We publish our implementation in https://anonymous.4open.science/r/NLPB-04A3 and provide the prompts in Appendix A.3 to further increase the reproducibility.

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

# A  FURTHER ANALYSIS

## A.1  EVALUATING TEXT RELEVANCE

**Table 5:** Relevance between LLM generated answers and ground-truth answers. We adopt BLEU, ROUGE-L, and CIDEr to represent the sentence relevance.

| Model | Setting | BLEU | | ROUGE-L | | CIDEr | |
|---|---|---|---|---|---|---|---|
| | | w/ Context | w/o Context | w/ Context | w/o Context | w/ Context | w/o Context |
| GPT-3.5 | ZS | 0.10 | 5.83 | 0.48 | 8.75 | 10.91 | 14.23 |
| | ZS+SYS | 0.11 | 5.20 | 5.04 | 8.69 | 7.75 | 0.00 |
| | ZS+CoT | 0.16 | 4.82 | 0.28 | 13.19 | 11.35 | 14.74 |
| | ZS+CoT+SYS | 0.47 | 5.28 | 0.23 | 13.94 | 10.08 | 3.79 |
| | ZS+ToT | - | - | - | 0.00 | 0.00 | 0.00 |
| | FS | 0.15 | 5.18 | 1.99 | 12.55 | 12.02 | 18.01 |
| | FS+SYS | 0.55 | 6.26 | 6.31 | 17.01 | 13.26 | 27.19 |
| | FS+CoT | 0.10 | 4.59 | 3.47 | 9.26 | 10.41 | 15.14 |
| | FS+SYS+CoT | 0.31 | 5.07 | 0.01 | 14.04 | 12.05 | 17.41 |
| | FS+ToT | - | - | - | 6.86 | 7.69 | 0.19 |
| GPT-4 | ZS | 0.63 | 6.47 | 9.32 | 11.83 | 9.85 | 6.28 |
| | ZS+SYS | 0.67 | 7.03 | 5.40 | 14.31 | 9.46 | 0.14 |
| | ZS+CoT | 1.12 | 7.00 | 5.05 | 10.68 | 9.67 | 25.16 |
| | ZS+CoT+SYS | 1.14 | 7.29 | 2.66 | 15.69 | 10.16 | 5.29 |
| | ZS+ToT | - | - | - | 0.00 | 0.00 | 0.00 |
| | FS | 1.34 | 7.76 | 15.24 | 17.09 | 11.57 | 5.59 |
| | FS+SYS | 2.00 | 9.94 | 21.85 | 20.17 | 14.32 | 15.90 |
| | FS+CoT | 0.71 | 6.48 | 7.71 | 13.77 | 11.13 | 3.09 |
| | FS+SYS+CoT | 0.90 | 6.82 | 7.87 | 17.93 | 14.98 | 35.73 |
| | FS+ToT | - | - | - | 15.35 | 10.62 | 9.21 |
| PaLM-2 | ZS | 3.35 | 10.89 | 23.19 | 23.21 | 14.06 | 19.02 |
| | ZS+SYS | 6.96 | 9.27 | 22.15 | 25.66 | 12.70 | 18.85 |
| | ZS+CoT | 3.05 | 9.31 | 11.66 | 15.30 | 11.71 | 14.36 |
| | ZS+CoT+SYS | 8.09 | 9.00 | 26.96 | 23.55 | 11.62 | 31.52 |
| | ZS+ToT | - | - | - | 0.00 | 0.00 | 0.00 |
| | FS | 1.16 | 13.28 | 57.25 | 26.74 | 13.68 | 17.67 |
| | FS+SYS | 4.03 | 9.47 | 28.31 | 24.60 | 15.62 | 32.05 |
| | FS+CoT | 0.33 | 8.19 | 20.83 | 14.32 | 9.86 | 4.68 |
| | FS+SYS+CoT | 1.82 | 9.60 | 24.63 | 15.00 | 8.99 | 9.57 |
| | FS+ToT | - | - | - | 0.50 | 0.72 | 1.89 |
| LLAMA-2 (13b) | ZS | 0.19 | 4.80 | 0.02 | 9.69 | 8.66 | 0.00 |
| | ZS+SYS | 0.37 | 5.02 | 0.00 | 11.35 | 9.64 | 1.21 |
| | ZS+CoT | 0.95 | 5.08 | 0.06 | 12.53 | 7.86 | 0.06 |
| | ZS+CoT+SYS | 1.23 | 5.46 | 0.16 | 12.89 | 7.34 | 1.09 |
| | ZS+ToT | - | - | - | - | - | - |
| | FS | - | - | - | 5.34 | 7.18 | 0.00 |
| | FS+SYS | - | - | - | 3.18 | 7.18 | 0.00 |
| | FS+CoT | - | - | - | 3.78 | 7.84 | 0.00 |
| | FS+SYS+CoT | - | - | - | 3.25 | 6.32 | 0.00 |
| | FS+ToT | - | - | - | - | - | - |
| LLAMA-2 (70b) | ZS | 0.10 | 4.96 | 0.00 | 6.47 | 8.14 | 5.57 |
| | ZS+SYS | 0.16 | 5.88 | 2.10 | 9.72 | 9.60 | 0.36 |
| | ZS+CoT | 0.91 | 5.05 | 0.46 | 13.73 | 7.51 | 1.24 |
| | ZS+CoT+SYS | 1.69 | 5.63 | 0.04 | 14.34 | 8.50 | 3.23 |
| | ZS+ToT | - | - | - | - | - | - |
| | FS | 0.02 | 4.04 | 0.00 | 4.82 | 7.88 | 0.53 |
| | FS+SYS | 0.08 | 4.81 | 0.01 | 8.71 | 8.85 | 3.13 |
| | FS+CoT | 0.08 | 3.17 | 0.00 | 4.62 | 8.63 | 2.03 |
| | FS+SYS+CoT | 0.16 | 3.40 | 0.00 | 5.54 | 8.22 | 0.00 |
| | FS+ToT | - | - | - | - | - | - |

Text relevance is a crucial metric, highlighting the relationship between two sentences and ensuring that a generated answer aligns with the task at hand. Classical metrics like BLEU and ROUGE-L measure the shared sequences between pairs of sentences: BLEU focuses on the n-gram overlap, while ROUGE-L captures the lengthiest common sequence. CIDEr refines the ROUGE-L metric by accounting for synonyms, word frequency, and scene graphs. We evaluated short-answer questions (with unique answers) generated by GPT-3.5, GPT-4, PaLM-2, and LLAMA-2 (13b and 70b) using the BLEU, ROUGE-L, and CIDEr metrics. Our collective findings are presented in Table 5. Interestingly, PaLM 2 displayed notably higher scores compared to other models but exhibited low accuracy, as seen in Table 3. Delving into the errors of PaLM 2, we discerned that, while it can provide accurate descriptions of specific concepts, it often muddles the logical connections between these concepts and redundantly reiterates irrelevant ones. An illustrative error from PaLM 2 is showcased in Figure 6, where the model erroneously repeats certain concepts. However, this repetition ironically leads to heightened text relevance scores. This observation underscores a limitation inherent in using text relevance metrics for evaluating LLMs.

```
Question:
Check the following four paragraphs:

1. Uppsala, in Sweden, will be the site of the 1969 International Conference on Computational Linguistics, September 1-4.
It is expected that some 250 linguists will attend from Asia, West Europe, East Europe including Russia, and the United
States.

2. Tempting as it may be, we shouldn't embrace every popular issue that comes along.

3. A well-groomed car reflects its owner. The car you drive says a lot about you.

4. But I don't think endorsing a specific nuclear freeze proposal is appropriate for CCC. We should limit our involvement
in defense and weaponry to matters of process, such as exposing the weapons industry's influence on the political process.

What rhetorical relations in RST they illustrate?
```

```
Ground truth:
1. Elaboration; 2. Concession; 3. Restatement; 4. Antithesis
```

```
PaLM 2:
1. Location, Elaboration; 2. Condition, Elaboration; 3. Elaboration, Elaboration; 4. Elaboration, Elaboration
```

**Figure 6:** Example of wrong answer generated by PaLM 2. It is obvious that PaLM 2 repeat some wrong concept many times, but this will significantly increase the relevance between ground truth and the generated answer.

```
Question:
Describe why diversity, grice's maxims and content selection that arise in language generation don't exist in language understanding
(parsing).
```

```
Ground truth:

1. Diversity: when generating material that talks about the same entity multiple times, it may be desirable to use multiple different ways
to refer to it.  For example, in news articles, a person is usually first referred to by their full name, then by their surname, and then
with a pronoun.
2. Grice's maxims: these maxims control how a dialogue agent should interact.  They prevent the agent from giving too little information,
too much information, false information, etc.
3. Content selection: Before even deciding how to say something, a generation system often needs to decide what to say.
```

```
Wrong Answer (GPT-3.5):

Diversity, Grice's maxims, and content selection do exist in language understanding (parsing).

Language understanding involves interpreting and making sense of the input language, which requires considering the various possible
interpretations and understanding the full range of meanings. This inherently involves diversity in understanding, as different individuals
may have different interpretations and perspectives on the same input.

Grice's maxims, proposed by philosopher H.P. Grice, are principles that guide effective communication, such as the maxim of quantity
(providing enough information) and the maxim of relevance (staying on topic). These maxims play a crucial role in language understanding as
they help in inferring intended meaning, resolving ambiguity, and constructing coherent interpretations.

When it comes to content selection in language understanding, the process involves identifying the relevant information from the input and
mapping it to the appropriate knowledge and context. This selection of content is essential for comprehending the meaning conveyed by the
input and building a coherent representation of the message or information.

Therefore, diversity, Grice's maxims, and content selection are integral aspects of language understanding, and they contribute to
accurately comprehending and interpreting the input language.
```

```
Wrong reason (LD, IA, LR):
The primary issue with the student's answer is a misalignment with the question's intent. The student defends the presence of these aspects
in language understanding rather than explaining why they don't arise in the same way they do in language generation. This indicates
potential misunderstandings in both the distinctions between generation and understanding and in the roles of the aspects in question.
```

**Figure 7:** An example of short answer error in GPT-3.5, where the answer of GPT-3.5 cannot align the question, indicating the lack of logical decomposition and analysis, identification of assumptions, and logical reasoning skills.

## A.2 ERROR SAMPLES

We provide some error samples generated by GPT-3.5 in Figure 7 and Figure 8 for a better understanding of the error reason in Section 4.2.

## A.3 PROMPT TEMPLATE

We designed specific prompts for each type of question, and we summarized those prompts in this section. Figure 9 shows the system prompt, Figure 10 shows the prompt template for multiple choice questions, Figure 11 shows the prompt template for the short answer, and Figure 12 shows the prompt template for math questions. {input} is the place for input questions, {thought} denotes the middle-process prompt used for CoT. We use a two-stage method for short answer questions,

```
Question:
A corpus contains 1,000,000 tokens, including 6,000 instances of the word "the". The next four most frequent words are "of", "and", "to",
and "a". What are the expected counts for each of these words in the corpus?
```

```
Ground truth:

"of": 6000/2=3000
"and": 6000/3=2000
"to": 6000/4=1500
"a": 6000/5=1200
```

```
Wrong Answer (GPT-3.5):

as follows:
- "of": (6000/1000000) * 1000000 = 6000
- "and": (6000/1000000) * 1000000 = 6000
- "to": (6000/1000000) * 1000000 = 6000
- "a": (6000/1000000) * 1000000 = 6000
```

```
Wrong reason (LD, PD, AR):
The LLM doesn't seem to grasp the concept of word frequencies and how they decrease in a typical corpus, which needs to understand the
basic principles of language distribution and frequency to address this type of problem correctly.
```

**Figure 8:** An example of a math error in GPT-3.5, where GPT-3.5 cannot understand the principles of language distribution and frequency, indicating the lack of logical decomposition and analysis, problem deduction, and abstract reasoning skills.

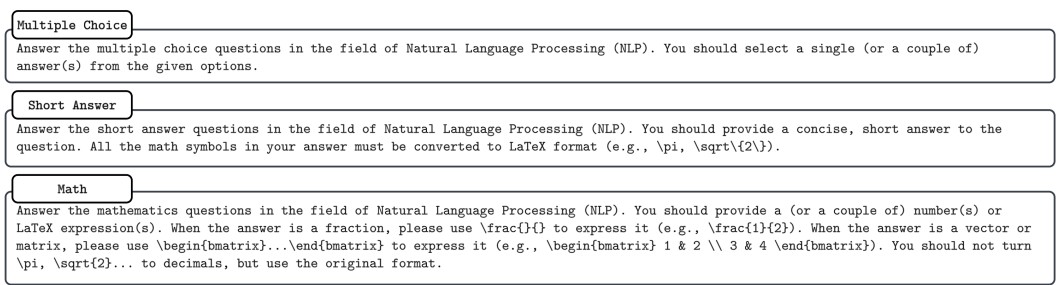

```
Multiple Choice
Answer the multiple choice questions in the field of Natural Language Processing (NLP). You should select a single (or a couple of)
answer(s) from the given options.
```

```
Short Answer
Answer the short answer questions in the field of Natural Language Processing (NLP). You should provide a concise, short answer to the
question. All the math symbols in your answer must be converted to LaTeX format (e.g., \pi, \sqrt\{2\}).
```

```
Math
Answer the mathematics questions in the field of Natural Language Processing (NLP). You should provide a (or a couple of) number(s) or
LaTeX expression(s). When the answer is a fraction, please use \frac{}{} to express it (e.g., \frac{1}{2}). When the answer is a vector or
matrix, please use \begin{bmatrix}...\end{bmatrix} to express it (e.g., \begin{bmatrix} 1 & 2 \\ 3 & 4 \end{bmatrix}). You should not turn
\pi, \sqrt{2}... to decimals, but use the original format.
```

**Figure 9:** System prompt for multiple choice, short answer, and math questions.

in which the thought is generated by the LLM itself, and a format template for multiple choice and math questions. Specifically in math, we put the problem-solving process into the CoT prompt ({process}) as the "thought". Note that the prompts listed here are all zero-shot prompts, and the few-shot prompt is based on the zero-shot by further adding some example questions.

## B  USER INTERFACE

The original dataset has a lot of handwriting scripts. We, therefore, create a UI interface to transform those handwriting scripts to JSON format manually. Figure 13 shows the screenshot of our UI interface. To ensure the correctness of input questions, we developed a real-time preview window for annotators to revise their input.

## C  COMPARISON BETWEEN PREVIOUS BENCHMARKS AND NLPBENCH

To clearly distinguish the difference between each benchmark, we summarized the characteristics of each benchmark in three dimensions: dataset composition, tested methods, and analysis methods. Table 6 shows the difference between each benchmark. Our dataset introduces the questions that require LLMs to answer with multi-turn communication and contains all types of questions that can test the LLMs' ability comprehensively.



**Prompt template for Multiple Choice Questions**

```
Answer the final multiple choice question.
Your output must be only numbered, splitting by commas
(e.g., 0,1,...) with no descriptions.
Example Input:
ChatGPT is created by which of the following companies?
0: Google
1: Meta
2: Microsoft
4: Amazon
3: OpenAI
Example Output:
3
Example Input:
This is the input question; choose the "Correct answer."
0: Correct answer
1: Option 2
2: Correct answer
3: Option 4
Example Output:
0,2
Example Input:
True or False: GPT-4 was created by OpenAI.
0: True
1: False
Example Output:
0
Input (You need to answer this question):
{input}
Output:
```

**Prompt template for Multiple Choice Questions (with CoT)**

```
Answer the final multiple choice question. Your output must be only
numbered, splitting by commas (e.g., 0,1,...) with no descriptions.
Example Input:
ChatGPT is created by which of the following companies?
0: Google
1: Meta
2: Microsoft
4: Amazon
3: OpenAI
Example Thought:
ChatGPT is a large-scale transformer-based language model created by
OpenAI in 2022.
Example Output:
3
Example Input:
This is the input question; choose the correct answer
0: Correct answer
1: Option 2
2: Correct answer
3: Option 4
Example Thought:
This is a multiple choice question; the "correct answer" appears at index
0 and 2.
Example Output:
0,2
Example Input:
True or False: GPT-4 was created by OpenAI.
0: True
1: False
Example Thought:
GPT-4 was created by OpenAI in 2022.
Example Output:
0
Input (You need to answer this question):
{input}
Output:
```



**Figure 10:** Zero-shot prompt template for multiple choice questions.



**Prompt template for Short Answer Questions**

```
Answer the following short answer question. Your answer
should be no more than 150 words.
Input (You need to answer this question):
{input}
Output:
```

**Prompt template for Short Answer Questions (with CoT)**

```
(Stage 1):
Answer the following short answer question. Your answer should be
no more than 150 words.
Input (You need to answer this question):
{input}
Let's think step by step.

(Stage 2):
Answer the following short answer question. Your answer should be
no more than 150 words.
Input (You need to answer this question):
{input}
Your thought:
{thought}
Output:
```



**Figure 11:** Zero-shot prompt template for short answer questions. Note that we use a two-stage method to generate the middle process for CoT.



**Prompt template for Math Questions**

```
Answer the following math question. Your answer should
be a number, a list of numbers, or a LaTeX expression.
Example Input:
1 + 1
Example Output:
2
Example Input:
\frac{1}{2} + \frac{1}{3}
Example Output:
\frac{5}{6}
Example Input:
f(x) = 4x^2 + 3y
Solve the \frac{\partial f(x)}{\partial x}
Example Output:
8x
Input (You need to answer this question):
{input}
Output:
```

**Prompt template for Math Questions (with CoT)**

```
Answer the following math question. Your answer should be a
number, a list of numbers, or a LaTeX expression.
Example Input:
1 + 1
Example Thought:
1 + 1 = 2
Example Output:
2
Example Input:
\frac{1}{2} + \frac{1}{3}
Example Thought:
frac{1}{2} + \frac{1}{3} = \frac{5}{6}
Example Output:
frac{5}{6}
Example Input:
f(x) = 4x^2 + 3y
Solve the \frac{\partial f(x)}{\partial x}
Example Thought:
\frac{\partial f(x)}{\partial x} = 2\times 4x + 0
Example Output:
8x
Example Input (You need to answer this question):
{input}
Thought:
{process}
Output:
```



**Figure 12:** Zero-shot prompt template for math questions. We input the middle process as the "thought" for CoT.

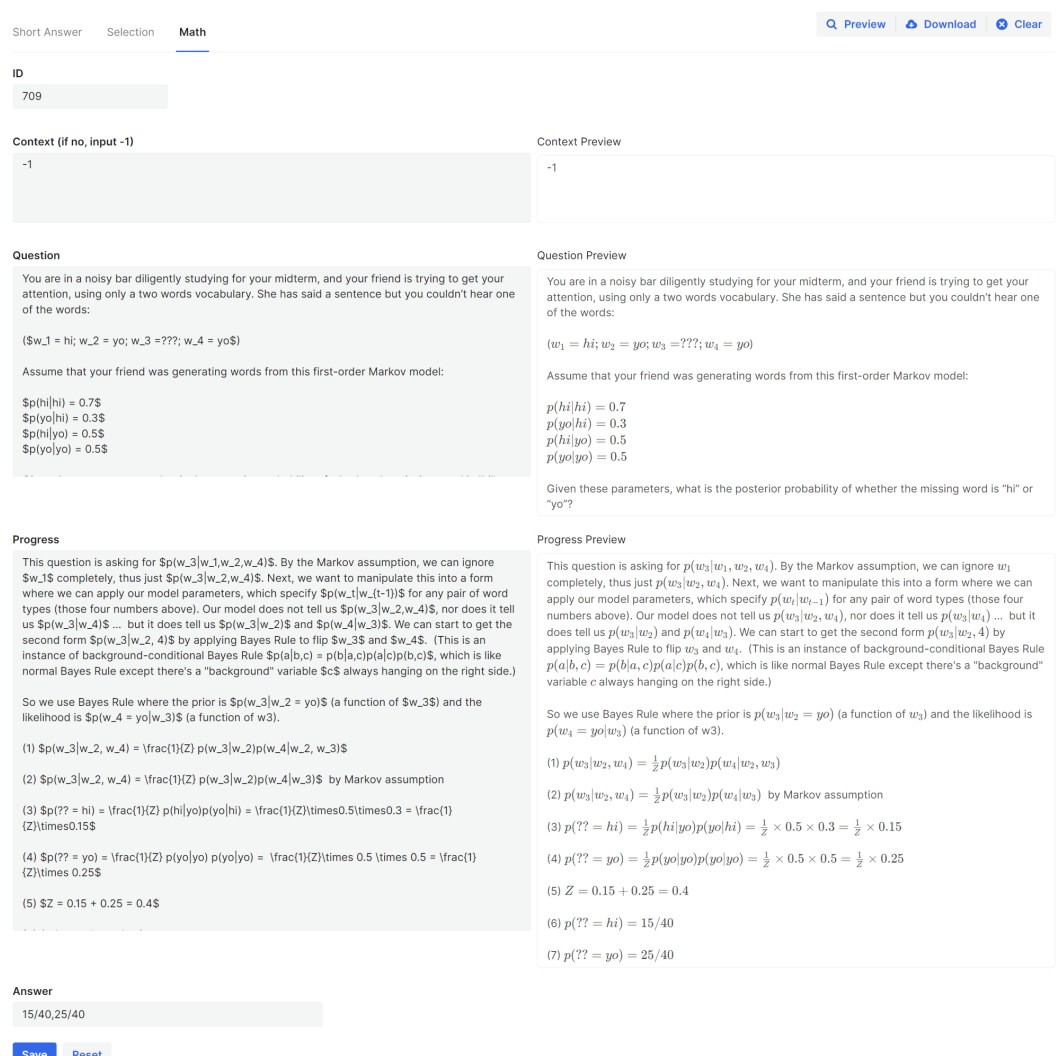

**Figure 13:** The UI design of data processing and annotation.

**Table 6:** Comparison of NLPBench with other benchmarks. "Level" represents the grade level of problems. "w/ Solution" represents whether problems contain detailed solutions. "Type" represents what format most problems of the dataset use. "AP" denotes whether the benchmark uses the advanced prompting strategies, "MC" denotes multiple-choice format, "MT" denotes the question requires answer in multi-turn communication, and "Free" denotes free-response format. "Human" indicates whether the analysis process employs a human annotation process. "Auto" represents whether the analysis process uses an automatic annotation process.

| Benchmark | Dataset | | | Experiment | | | | Analysis | |
| | Level | w/ Solution | Type | ZS | FS | AP | MT | Human | Auto |
| --- | --- | --- | --- | --- | --- | --- | --- | --- | --- |
| ScienceQA (Lu et al., 2022) | Grade 1-12 | Yes | MC | Yes | Yes | Yes | No | No | No |
| IconQA (Lu et al., 2021b) | Grade 1-12 | No | MC | No | Yes | No | No | No | No |
| TabMWP (Lu et al., 2023a) | Grade 1-12 | Yes | Free | No | Yes | No | No | No | No |
| GSM8K (Cobbe et al., 2021) | Grade 1-12 | Yes | Free | No | Yes | No | No | No | No |
| MATH (Hendrycks et al., 2021) | High School | Yes | Free | No | Yes | No | No | No | No |
| LILA (Mishra et al., 2022) | High School | Yes | Free | Yes | Yes | No | No | No | No |
| MNLU (Hendrycks et al., 2020) | High School & College | No | MC | No | Yes | No | No | No | No |
| CEval (Huang et al., 2023) | High School & College | No | MC | No | Yes | Yes | No | No | No |
| AGIEval (Zhong et al., 2023) | High School & College | No | MC | Yes | Yes | Yes | No | No | No |
| TheroemQA (Chen et al., 2023) | College | No | Free | No | Yes | Yes | No | No | No |
| SciBench (Wang et al., 2023) | College | Yes | Free | Yes | Yes | Yes | No | Yes | Yes |
| NLPBench | College | Yes | Free & MC | Yes | Yes | Yes | Yes | Yes | Yes |

