# OpenReview forum: "NLPBench: Evaluating Large Language Models on Solving NLP Problems"
_ICLR.cc/2024/Conference — Submitted to ICLR 2024_

### Official Review · Reviewer_tbcE · 2023-10-31

**Soundness:** 3 good
**Presentation:** 3 good
**Contribution:** 3 good
**Rating:** 5
**Confidence:** 4

**Summary:**

The authors develop a new benchmark dataset (NLPBench) to evaluate the large language models for solving NLP problems. NLPBench comprises 378 college-level NLP questions (with and without context) spanning various NLP topics sourced from some University's prior final exams. NLPBench was evaluated on different LLMs such as GPT-4, PaLM-2, and LLAMA-2 using advanced prompting strategies including chain-of-thought (CoT) and tree-of-thought (ToT) and different decoding strategies such as self-consistency. The results show that NLPBench illuminates specific shortcomings in LLMs’ scientific problem-solving skills, with weaknesses in logical decomposition and reasoning notably affecting performance.

**Strengths:**

1. Introduces a new dataset that challenges the current state-of-the-art prompting strategies and which is useful in evaluating the performance of LLMs.

2. The paper is well-written and easy to follow.

3. The authors carried out extensive experiments and evaluations that included recent prompting approaches and LLMs.

**Weaknesses:**

In general, NLPBench is useful, but I think there are some clarities on how it was collected that are missing:

1. How did you get access to the final exams included in the dataset?  How do you ensure that these exams were not already online and that some recent models like GPT-4 have already included them in their training dataset? It was not clear from the paper how you checked that.  You mentioned that you curate questions that are not readily accessible online and couldn’t be easily extracted or transformed into text, but how do you measure that specific content cannot be easily extracted or transformed into text? what reliable techniques did you use to do that?

2. In what years these final exams were given? This information might help in the development of future related datasets.

3. In the data selection process, you mentioned that you selected 400 questions out of 1000 total questions, what were the criteria? Was it a random selection?

4. The dataset size looks small, why in Table 6, was not there any comparison for the NLPBench size to other benchmarks?

5. The very closely related benchmark (SciBench) also includes math problems, did you check if NLPBench's math-related problems are not also in SciBench? I did not see this evaluation in the paper.

6. How many expert human annotators were involved in the dataset processing?

**Minor:**
Some typos:
- Page 3 in the paragraph that describes the complex structure: multi-tern -> multi-tern
- Page 4: zero-shot and few-shot(FS+ToT, FS+ToT) -> (ZS+ToT, FS+ToT)

**Questions:**

- How many universities were included in the data collection?
- For other questions, please check the weakness section.

---

> ### Author Response · Authors · 2023-11-13
> **Author's Response to Reviewer tbcE**
>
> Thank you for your valuable and meaningful feedback and for pointing out typos in our work; we updated our manuscript, reflected reviews and fixed typos, and carefully addressed your questions and concerns below.
>
> > **How did you get access to the dataset? How did you ensure the dataset has not been included in LLM? In what years these final exams were given?**
>
> We collected 1,000 NLP exam questions over a decade, from 2013 to 2023. For the integrity and fairness of the final exams, these questions have not been available online, so they cannot be part of other datasets, and there is no risk of training data contamination. Professors and teaching assistants (TAs) contributed new questions to this repository each semester. This extensive set comprises three types of questions: 1) Online-sourced questions, refined by TAs to differentiate them from their original versions, ensuring their uniqueness for final exams, along with a thorough verification of answers. 2) Original questions formulated by professors, drawing from their teaching experience. 3) Original questions developed by TAs. Over 30 TAs have been involved in the completion of this dataset. We have made this clear in the updated version.
>
> > **What were the criteria for data selection?**
>
> In the raw data, many questions include figures and tables in the context. We discard those questions since we cannot easily translate them into text formats like images and tables. We have made this clear in the updated version. Considering that multimodal LLM (like GPT-4V) has been announced after this work, we will add more experiments of multimodal models in our future work.
>
> > **Why not compare dataset size?**
>
> A good benchmark dataset should be able to reveal the differences among LLMs through their performance. According to Table 3, Figures 2 and 3, NLPBench can achieve this with the current amount of questions. We also provided in-depth error analyses in Section 4. Therefore, we did not focus on tweaking the number of questions in each dataset but more on the features (e.g., new types of questions or new evaluation processes).
>
> > **Did you check if NLPBench's math-related problems are not also in SciBench?**
>
> SciBench extracts questions from ten textbooks in the fields of physics, chemistry, and pure math, while the questions in NLPBench are created by professors and TAs. Therefore, there is no overlap between SciBench and NLPBench.
>
> > **How many human annotators are involved?**
>
> The main contributors to the original dataset are at least 30 TAs, each of them completing part of the questions’ or answers’ completion. Furthermore, three human experts are involved in the data processing stage: one for transiting the original format into the text format with a front-end interface and two for checking the results. It took about one week to complete the development of the interface and dataset transformation. Two of them were further involved in the free-response answer evaluation and error analysis stage. Evaluating each answer needs ~5 minutes per annotator, and it took about 13 hours per annotator to finish the evaluation.

---

### Official Review · Reviewer_FuWu · 2023-10-31

**Soundness:** 2 fair
**Presentation:** 2 fair
**Contribution:** 2 fair
**Rating:** 5
**Confidence:** 3

**Summary:**

The authors propose a new LLM benchmarking dataset consisting of 378 questions collected from Universities’ NLP final exams with short-answers,  multiple-choice questions, math questions with step-by-step solutions. Some questions come with a context information. Different LLMs (GPT-3.5/4, PaLM-2, and LLAMA-2) are benchmarked on the dataset with different prompting strategies with a combination of zero-shot, few-shot and chain of thought (CoT) and tree of thought (ToT). The experimental results show that prompting strategies such as can reduce performance, in particular for small size models. The evaluation shows LLM limitations in reasoning, logical decomposition.

**Strengths:**

- This work aim to expand the scope of LLM benchmarking and address new perspectives by contributing new datasets and benchmarking scenarios
- The evaluation includes both close and open models. It is important to understand the gap between these model types

**Weaknesses:**

- The size of the dataset is relatively small and might limit the conclusion drawn from the results.
- The gap filled by the proposed by introducing NLPBench (Table 6) is rather narrow and needs stronger justification. It covers a broader context than the claimed main focus of NLP-related topics such as Math.
- The analysis conclusion are mostly known, such as small LLMs have inconsistent results with advanced prompting. LLM limitation on problem-solving tasks.

**Questions:**

- How open questions have been evaluated ? The paper mentions only human annotation of errors.
- Under Inaccessibility, details are missing. How did you make sure that the questions are not part of other datasets ? What are risks of contamination by LLM training data  ?
- I expected that the top-3 models is including PalM but in 4.2, Llama-2 was taken instead. Is there a justification for this choice ?

---

> ### Author Response · Authors · 2023-11-13
> **Author's Response to Reviewer FuWu**
>
> Thank you for your valuable and meaningful feedback on our work! We carefully and thoroughly address your questions and concerns below.
>
> > **NLPBench covers a broader context than the claimed main focus of NLP-related topics such as Math.**
>
> Unlike pure math, math problems in the NLP domain are closer to applications. For example, the language generation models are constructed based on the conditional probability. The context of the math problem in NLPBench is strongly correlated with the NLP context (see examples in Figure 1). Therefore, in general, NLPBench covers different aspects of  NLP-related topics, including related math questions.
>
> > **The analysis conclusions are mostly known.**
>
> To the best of our knowledge, this is the first LLM benchmark in the domain of NLP, which not only verifies several long-discussed conjectures but also shows several novel discoveries that have not been explored by other works by the time we submit. For example, (1) The failures of chain-of-thought and tree-of-thought on 13B-sized, 70B-sized, or even larger models, which set up a boundary for the application of advanced prompting strategies; (2) We also discovered a significant drawback in logical thinking skills, which can be further used to, for example, explain why training on some specific resources (like code) will improve the performance on others because many logical symbols in those resources may be involved in the construction of logical thinking skills and can also be used to instruct the pre-training and post-training stage of LLMs.
>
> > **How open questions have been evaluated?**
>
> We first prompt a GPT-4 to evaluate the free-response questions’ answers, and two human domain experts further read through and carefully provide feedback. On average, for each question, they need ~5 minutes; it took about 13 hours per annotator to finish the work. They evaluated answers in the following procedure: (1) figure out the scoring point from the ground truth answer, and (2) compare the generated answer with those scoring points at the semantic level.
>
> > **How did you make sure that the questions are not part of other datasets?**
>
> For the integrity and fairness of the final exams, these questions are made originally or refined by professors and teaching assistants and unavailable online, so they cannot be part of other datasets, and there is no risk of contamination of training data.
>
> > **Is there a justification for the choice of GPT-3.5/4 and LLAMA-2 (70b)?**
>
> We included LLAMA-2 (70b) in the error analysis because LLAMA-2 (70b) is an open-source LLM, which will make our evaluation more trackable and will not influenced by the online API update. On the other hand, we chose GPT-3.5/4 because (1) it is one of the most common online API LLMs; and (2) unlike PaLM, its API is accessible to all users.

---

### Official Review · Reviewer_YrAB · 2023-11-01

**Soundness:** 3 good
**Presentation:** 3 good
**Contribution:** 2 fair
**Rating:** 6
**Confidence:** 4

**Summary:**

This paper presents a benchmark dataset for NLP-related questions for LLMs to answer. The collected dataset, NLPBench, consists of 378 college-level NLP questions and various LLMs are tested on this benchmark. In addition, this paper also compares different prompting strategies, but finds out that the results are not consistent with the "advancement" of the technique.

**Strengths:**

* Interesting angle and scope for LLM evaluation
* Experiments of the effectiveness of different prompting strategies under this task

**Weaknesses:**

* The scope of the evaluation is quite limited
* The dataset is small and relatively the evaluation cost is high

**Questions:**

* Could this approach be adapted to other areas or domains? Would be nice to discuss about it and also the transfer cost.

---

> ### Author Response · Authors · 2023-11-13
> **Author's Response to Reviewer YrAB**
>
> Thank you for your valuable and constructive feedback! We carefully and thoroughly address your questions and concerns below.
>
> > **Could this approach be adapted to other areas or domains? It would be nice to discuss about it and also the transfer cost.**
>
> Our automated evaluation can be directly adapted to other domains. We provided a complete codebase, which only requires minor changes for the new domain (e.g., rewriting the prompt and a JSON formatted dataset) to complete automated evaluations and obtain results like Table 3, 4, and 5. The human evaluation (like in Figure 5 and the evaluation of free-response answers) can be costly, considering that the domain-specific evaluations may require experienced domain experts and cannot be approached automatically. We tried our best to reduce the evaluation cost by involving GPT-4 as an assistant and caching the result with the same parameters by leveraging AutoGen.

---

### Author Response · Authors · 2023-11-13
**Revision Description and General Response**

We would like to thank all reviewers for their constructive comments! We uploaded a revision of our manuscript based on the review. Specifically, we made the following changes:

- We polished the abstract with a detailed description of the source of our dataset.
- Section 2 (Data selection) is now enriched with detailed information on the dataset source.
- We shortened the conclusion with more concise information about the whole manuscript.
- We fixed the typo mentioned by Reviewer `tbcE`.

**We also want to clarify the scale of the dataset below:**:

Our dataset comprises contributions from professors and more than 30 teaching assistants as part of the human labor force required for completion. We plan to expand the dataset's volume; however, this expansion is a time-intensive process. As the first benchmark for evaluating NLP problem-solving, our dataset's present magnitude is sufficient to expose the shortcomings of large language models (LLMs) and to discern the variances in their responses to NLP-related questions, and the proposed NLPBench took the first step towards the goal.

---

### Author Response · Authors · 2023-11-21
**We've carefully addressed each of the questions you raised and hoping for your valuable feedback**

Dear Reviewer `YrAB`, `FuWu`, and `tbcE`,

We greatly appreciate your thoughtful and insightful review. Your question has been carefully considered, and we have made every effort to provide thorough responses. We cordially invite you to review our answers in the hope that they align with your own insights. Your investment of time and expertise in evaluating our work has had a profound impact on us.

Sincerely,

Authors

---

### Meta-Review · Area_Chair_LXyp · 2023-12-20

**Metareview:**

This paper presents a dataset of final exam problems and answers in the university classes of natural language processing (NLP). While. this must have taken a lot of work and the resulting dataset is useful, it is not sufficient to warrant a publication at ICLR. For it to have scientific contribution, I would like to see an in-depth discussion about how the NLP problems and answers are different, from an LLM perspective, from the physics and other science questions in the SciBench. Without that discussion, it is just adding another dataset. Granted, benchmark datasets are necessary in advancing the field, but to be a full paper at ICLR, I think more insights about the dataset is needed, so that future researchers can learn from this experience in building more diverse and unique benchmark datasets.

**Justification For Why Not Higher Score:**

There is a lack of discussion about how NLPBench differs from SciBench, other than just the topic.

**Justification For Why Not Lower Score:**

N/A

---

### Decision · Program_Chairs · 2024-01-16

Reject